# Causal Effect of the Tokyo 2020 Olympic and Paralympic Games on the Number of COVID-19 Cases under COVID-19 Pandemic: An Ecological Study Using the Synthetic Control Method

**DOI:** 10.3390/jpm12020209

**Published:** 2022-02-03

**Authors:** Norio Yamamoto, Toshiharu Mitsuhashi, Yuuki Tsuchihashi, Takashi Yorifuji

**Affiliations:** 1Department of Epidemiology, Graduate School of Medicine, Dentistry and Pharmaceutical Sciences, Okayama University, Okayama 700-8558, Japan; yorichan@md.okayama-u.ac.jp; 2Department of Orthopedic Surgery, Miyamoto Orthopedic Hospital, Okayama 773-8236, Japan; 3Systematic Review Workshop Peer Support Group (SRWS-PSG), Osaka 541-0043, Japan; 4Center for Innovative Clinical Medicine, Okayama University Hospital, Okayama 541-0043, Japan; mitsuh-t@cc.okayama-u.ac.jp; 5Center for Field Epidemic Intelligence, Research, and Professional Development, National Institute of Infectious Diseases, Tokyo 162-8640, Japan; yuuki@niid.go.jp

**Keywords:** Tokyo 2020, Olympic and Paralympic Games, COVID-19, pandemic, synthetic control method, causal effect, ecological study

## Abstract

Previous studies have not assessed the causal effect of the Olympic Games on the spread of pandemics. Using the synthetic control method and the national public city data in Japan recorded from February to September 2021, we estimated the causal effects of the Tokyo 2020 Olympic and Paralympic Games on the number of coronavirus disease 2019 (COVID-19) cases. The difference between the number of COVID-19 cases in Tokyo and a counterfactual “synthetic Tokyo” (created using synthetic control method) after the opening of the Tokyo 2020 Games (23 July 2021) widened gradually and then considerably over time. It was predicted that the Tokyo 2020 Games increased the number of COVID-19 cases in Tokyo by approximately 469.4 per 100,000 population from the opening of the event to 30 September. However, sensitivity analysis of the ratio of the pre- and post-game root mean square prediction errors using regression weights did not suggest robustness. Our results showed that the Tokyo 2020 Games probably increased the number of COVID-19 cases even under preventive regulations; however, the extent of this increase was difficult to estimate clearly due to an overlap with the fifth wave associated with the Delta variant.

## 1. Introduction

The Olympic Games are a large-scale international sporting event, wherein many international competitors and spectators visit the host country for a limited period. In the past, the Olympic Games have been conducted during other pandemics; examples include the 1920 Antwerp Olympics (held during the Spanish flu pandemic) and the 2016 Rio Olympics (held during the Zika virus outbreak in Brazil). In 2021, the Tokyo 2020 Olympics and the Tokyo 2020 Paralympic Games were held in Japan during the coronavirus disease 2019 (COVID-19) pandemic [1]. During the Olympic Games, each host country needs to delegate public health resources to the event. Because pandemics cause severe medical and financial difficulties throughout the world, the decision to host the Olympics under a pandemic is an important political judgement. However, studies have not systematically analyzed the association between the Olympic Games and the spread of infectious diseases.

Many researchers have attempted to objectively evaluate the various public health policies that have been adopted to curb the spread of COVID-19. The synthetic control method (SCM) is a method for policy evaluation [2,3]. In this method, a counterfactual trend is created after considering the time trends and the demographic and geographic disparities; it is then compared with the actual trend to evaluate the effectiveness of the policy. Some policies under the COVID-19 pandemic, such as the implementation of lockdowns, use of face masks, compulsory home quarantining, and mandatory COVID-19 vaccination certificates, have been evaluated using the SCM [2,3,4,5,6,7].

Estimating the influence of the Olympic Games on pandemics will provide important insights that will enable future decisions on infection control policies and hosting the event. Therefore, this study aimed to clarify the causal effect of the Tokyo 2020 Olympic Games on the number of COVID-19 cases using the SCM. We hypothesized that the number of cases of COVID-19 would increase due to the increased flow of people associated with the Olympic Games and the decreased distribution of public health resources to COVID-19 control.

## 2. Materials and Methods

This ecological study was performed using national public city data in Japan. We used SCM to clarify the causal effect of the Tokyo 2020 Games on the number of COVID-19 cases [2,8,9,10]. In SCM, the counterfactual outcome of the Tokyo 2020 Games not being held was estimated from a weighted combination of donor pools. This study was performed in accordance with the Declaration of Helsinki. It did not require ethical approval, because it analyzed anonymous public data and not individual personal data.

### 2.1. Situation in Japan before the Tokyo 2020 Games

In 2021, the Japanese government urged the public to avoid the “three Cs”, namely crowded places, closed spaces, and close-contact settings. It also temporarily declared “a state of emergency” (not a lockdown) in prefectures associated with a continuing surge of COVID-19 cases. Before the Tokyo 2020 Games, approximately 25% of the Japanese population had been vaccinated by June 2021 [11]. The impact of COVID-19 in Japan was relatively low; a cumulative 859,056 cases of infection and 15,110 deaths had been recorded in a total population of 125 million by 23 July 2021 [12]. On 23 July 2021, Japan faced a strong “fifth wave” of COVID-19. Some coronavirus variants, such as the highly transmissive Delta variant, have been spreading in Japan [13]. Globally, various infectious variants of COVID-19 are spreading [14]. 

### 2.2. Situation during the Tokyo 2020 Games

After a one-year postponement, Tokyo hosted the Tokyo 2020 Olympics from 23 July to 8 August 2021 and the Tokyo 2020 Paralympic Games from 24 August to 5 September 2021. Approximately 20,000 athletes and staff from more than 200 countries participated in the events [15]. Therefore, some infection control strategies were implemented. Games without spectators were held in the prefectures around Tokyo and Hokkaido. Most athletes and staff members had been vaccinated and were frequently tested for COVID-19 [16]. In accordance with the Olympic bubble system, they were restricted from moving outside the hotel and practice venue [16].

### 2.3. Study Period

This study was performed from 1 February to 30 September 2021. 1 February 2021 was defined as the beginning of the study period because the fourth wave of COVID-19 in Japan had started sometime between February and March 2021. The time lag between virus exposure and COVID-19 occurrence is estimated to be 1–2 weeks [15,17]. Therefore, 30 September 2021 was defined as the end of the study period because the Paralympics ended on 5 September and a few weeks had passed by then.

### 2.4. The Donor Pool

Abadie et al. defined donor pools as regions with similar characteristics to the region exposed to the event of interest [2]. We constructed the counterfactual “synthetic Tokyo” from a unit in the donor pool of prefectures in Japan. In terms of the influence of the Olympics-related people flow into Tokyo from abroad, the inclusion criteria for the donor pool mainly comprised the following: (1) distance from the host city (Tokyo) and the game venue and (2) presence of an advanced training camp. Advanced training camps for the Olympics or Paralympics were held throughout Japan. Therefore, based on the distance, we selected 29 prefectures west of Toyama, Gifu, and Aichi as the donor pool to attenuate the effect of the Olympic or Paralympic Games (Figure 1).

### 2.5. Outcomes

The primary study outcome was the daily number of newly confirmed COVID-19 cases in the Tokyo metropolitan area and in synthetic Tokyo after the Tokyo 2020 Olympic and Paralympic Games. These patients were diagnosed and publicly announced in Japan [12]. However, the patients’ identities and information on whether this was their first or second infection episode were unknown. The 7-day moving average of the new cases was evaluated as the smallest unit.

### 2.6. Variables

The social and economic variables analyzed for each prefecture comprised the proportion of the population over 65 years of age, proportion of the population aged between 15 and 64 years in 2019 [18], daily number of COVID-19 cases before the Tokyo 2020 Games [12], and proportion of the population vaccinated with the second dose on 22 July 2021 [11]. To determine the proportion of the variants of severe acute respiratory syndrome coronavirus 2 in circulation from 19 July to 25 July 2021, we used polymerase chain reaction (PCR) assays and the following formula: (number of cases of the L452R mutation (Delta variant) the PCR test was positive for)/(number of PCR tests performed for the L452R mutation) [19]. 

### 2.7. Statistical Analysis

We evaluated the causal effect of the Tokyo 2020 Games on the number of COVID-19 cases using the SCM, as proposed by Abadie et al. [2]. We calculated the weight of the no-event (control) region from the donor pool in a way that the average pre-event trend and other selected variables were similar to the trend and characteristics of the event region (Tokyo). The weights were assigned based on the extent of the donor pool’s similarity to Tokyo prior to the Tokyo 2020 Games. The weights from the donor pool were chosen as follows: the weights for each prefecture ranged from 0 to 1, such that the weights for all control prefectures totaled 1.

The following variables were used to determine the weights: number of new cases from 1 February to 22 July 2021, proportion of vaccinated individuals on 26 July 2021, proportion of the variants detected from 19 July to 25 July 2021, proportion of the population aged 15–64 years, and proportion of the population aged 65 years and above. In total, weights were calculated using 12 variables across 29 donor pools. The study period was divided into the following eight sections based on the trends of the 7-day moving average of the COVID-19 cases in Tokyo: end of the data 1 February, 22 July (the day before the opening ceremony of Tokyo 2020), the nadir of the decreasing trend (valley) 8 March, 15 June, the top of the increasing trend—13 May, middle days of the valleys—19 February, 10 April, 30 May, 4 July (Appendix A). The weights for the donor pool were chosen as follows. X1 is the (12 × 1) vector representing the variables of the Kanto region before the intervention; X0 is the (12 × 29) matrix signifying the variables of the control pool before the intervention; and W is the (29 × 1) vector denoting the weight of the control pool. The weight W = (w1, w2, …, w29) was selected to minimize ||X1-X0W||, subject to 0 ≦ wj ≦ 1 for all j, and w1 + w2 + … + w29 = 1 (Appendix A).

In addition, the SCM was performed by using regression weights [8]. This approach employs a linear combination for which the sum of the weights in the control pool is 1. Therefore, the weights were calculated as:W_reg_ = X_0_′(X_0×0_′)^−1^X_1_

In this approach, the weights ranged from less than zero to greater than one; this was different from the range of the synthetic control weights. While this allows us to create well-fitting synthetic controls, it may lead to extrapolation outside of the support of the data.

The causal effect was evaluated by the difference in the 7-day moving average of the number of new cases after the intervention (the Tokyo 2020 Games) between Tokyo and synthetic Tokyo, as shown in the graph. The root mean square prediction error (RMSPE) was calculated to evaluate the goodness of the weights.

Robustness was checked using the in-space placebo effect. The in-space placebo effect was calculated by considering each prefecture in the donor pools as if it were the intervention area. This placebo effect expresses the variability of the results under the null hypothesis. The ratio of the post-RMSPE to pre-RMSPE was calculated as the effect size; by comparing the ratio of Tokyo with those of the other prefectures, we could determine whether the effect size of Tokyo was larger than the null hypothesis. In the regression weights, the post-RMSPE-to-pre-RMSPE ratio could not be calculated because the pre-RMSPE was zero. Therefore, post-RMSPE was used for the comparison.

Categorical variables are expressed as numbers and percentages, while continuous variables are expressed as means and standard deviations. There were no missing data in this study. The analysis was performed using Stata/MP4 17.0 with the user-generated command “synth” 0.0.7 [20]. This calculation for regression weights was performed using Python 3.7.4, with NumPy 1.18.1, Pandas 0.25.3, and Jupyter Notebook 6.0.2. 

## 3. Results 

Table 1 displays the synthetic control and regression weights of each prefecture in Tokyo. The synthetic control weights indicated that a combination of the three prefectures, namely Ishikawa, Osaka, and Okinawa (weights: 0.455, 0.277, and 0.267, respectively), reproduced the COVID-19 case trends in Tokyo before the Tokyo 2020 Games. 

Table 2 describes the demographic data of the synthetic control estimators before the Tokyo 2020 Games. Most variables were adjusted to be well-balanced. The discrepancy in the COVID-19 cases between Tokyo and synthetic Tokyo from 4 June to 22 July 2021, and in the proportion of COVID-19 variants remains. The RMSPE of synthetic Tokyo (1.92) was less than that of the population-weighted average of the control pool (2.39). The regression weight of synthetic Tokyo was a perfect match for that of Tokyo.

The COVID-19 case trend in synthetic Tokyo, estimated using synthetic control weights, closely tracked the COVID-19 case trend in Tokyo during the entire period preceding the Tokyo 2020 Olympic and Paralympic Games (Figure 2 and Figure 3). Although the ideal setting was that the difference was zero before the opening of the Olympic Games, the difference was not regulated to zero before the opening of the Olympic Games (Figure 3). The difference in the COVID-19 cases between Tokyo and synthetic Tokyo after the Olympics were opened indicates the influence of the Tokyo 2020 Games on the number of COVID-19 cases. Soon after the 2020 Tokyo Olympics were opened, the gap between the two trend lines widened gradually and then considerably over time. The number of COVID-19 cases in Tokyo increased sharply, whereas that in synthetic Tokyo increased moderately. The largest difference in the number of COVID-19 cases between the two was 13.2 per 100,000 population; this was observed on 19 August 2021 (27 days after the opening of the Olympics; Figure 3). After the closing of the Paralympics, the difference sharply became zero. Thus, the number of COVID-19 cases in Tokyo, from the opening of the Tokyo 2020 Games to their closing on 30 September was predicted to have increased by approximately 469.4 cases per 100,000 population (Figure 4 and Figure 5). 

Similarly, the COVID-19 case trend in synthetic Tokyo, determined using regression weight, closely tracks the COVID-19 case trend in Tokyo throughout the period preceding the Tokyo 2020 Olympic and Paralympic Games (Figure 6 and Figure 7). Before the opening of the Olympics, the difference was regulated to approximately zero (Figure 7). Soon after the Tokyo 2020 Olympics were opened, the number of COVID-19 cases in synthetic Tokyo increased, but decreased between 11 August and 31 August 2021. Therefore, the number of COVID-19 cases in Tokyo, from the opening of the Tokyo 2020 Games to their closing on 30 September was predicted to have increased by approximately 70.3 per 100,000 population (Figure 8 and Figure 9). 

In the analysis for robustness, Tokyo ranked second among 30 prefectures in terms of the post-RMSPE determined using regression weight; this implies a large change in the donor pools (Appendix A, Figure 10). Conversely, Tokyo ranked 15th among 30 prefectures in terms of the post-RMSPE-to-pre-RMSPE ratio determined using synthetic control weight; this implied that a significant change did not occur among the donor pools. These findings suggest that the estimates of the causal effects of the Tokyo 2020 Games on the number of COVID-19 cases during the event may not be robust.

## 4. Discussion

This study estimated the causal effects of the Tokyo 2020 Games on the number of COVID-19 cases using SCM. Our findings indicated that the Tokyo 2020 Games probably increased the number of COVID-19 cases, despite the implementation of preventive regulations. However, the extent of the increase was difficult to estimate clearly, because it coincided with the fifth wave associated with the Delta variant (which began around early July).

We believe that the Tokyo 2020 Games would have contributed to an increase in the number of COVID-19 cases in Tokyo to some extent. In this study, SCM performed using synthetic control weights was more reliable than SCM performed using regression weights, because it provided a smoother curve after the Tokyo 2020 Games. When regression weights were used, the number of cases in synthetic Tokyo decreased between 11 August and 31 August 2021; however, this decrease was not observed in any other factual prefectures. We speculate that this decrease was merely because the case numbers were calculated using negative values of the weights. Therefore, we may have overestimated the results of the regression weight analysis. When SCM was performed using synthetic control weights, a steep increase in the number of infections was observed 1–2 weeks after the opening of the Olympics; this may have coincided with the influence of the Olympics and the onset of COVID-19. However, the impact of the Tokyo 2020 Games remained uncertain, because the in-space placebo effect was not extremely large in Tokyo as compared to in the prefectures in the donor pool. 

There are three possible explanations for our results regarding the comprehensive impact of the Tokyo 2020 Games. First, hosting the Olympic Games during the pandemic conveyed mixed messages to the public. The “Olympic mood” may have made it easy to neglect infection control practices. For example, a significant increase in the number of COVID-19 cases was observed 14 days after sporting events without strict face-mask requirements [21]. The behavior surrounding attendance at related events, rather than match attendance, may have uniquely contributed to the number of COVID-19 cases [22]. This may be epidemiologically plausible, because sporting events with mass gatherings make people enthusiastic; this decreases adherence to preventive regulations. Second, due to the relatively decreased delivery of messages on COVID-19 prevention during the Olympics, we believe that the public may not have been adequately informed of the preventive regulations. Third, the distribution of resources, including political and healthcare resources, also decreases during the event. Experience gained from the management of previous mass gathering events during the pandemic should have been applied here [21,22]. 

### 4.1. Strengths

This is the first study to analyze the causal effects of the Tokyo 2020 Games on the number of COVID-19 cases using SCM. By employing SCM, we were able to analyze the change in the number of COVID-19 cases over time and match the trends in Tokyo and synthetic Tokyo before the event [2,23]. Furthermore, the donor pool could be systematically selected by calculating the weights [8]. This could address the problem that the comparison groups were not sufficiently similar to each other in characteristics other than the intervention. Finally, this study had transparency and validity, because we used national public data. 

The results of this study can offer important insights for future decisions on hosting the Olympics and international mass gathering events and framing policies during pandemics, considering the socioeconomic benefits and burdens. This timely research on COVID-19 also provides a generalizable analytical framework for epidemiological evaluation. 

### 4.2. Limitations

This study has several limitations. First, the variables were not easy to adjust, because Tokyo is a megacity in Japan and has extreme values for some variables. In particular, the percentages of the variant strains and COVID-19 cases before the opening of the Tokyo 2020 Games could not be adjusted. The proportion of the Delta variant in the COVID-19 cases in Tokyo rapidly increased from 21.5% (28 June to 4 July 2021) to 94.0% (23 to 29 August 2021) [24,25]. In synthetic Tokyo, the proportion of variant strains detected was small; had it been high, the number of COVID-19 cases could have been higher than the calculated results in this study. Thus, we might have overestimated the causal effects of the Tokyo 2020 Games. Second, the impact of the Tokyo 2020 Games may have also been present in the prefectures of the donor pool. We selected 29 prefectures west of Aichi based on their distance from Tokyo. However, it is possible that these prefectures had been affected by Olympic-related factors, such as the presence of advanced training camps and flow of people from overseas, leading to an overestimation of the number of COVID-19 cases in synthetic Tokyo. Japan is a small island nation, and it was challenging to select a donor that would be completely uninfluenced by the Tokyo 2020 Games. For this reason, the causal effect of the Tokyo 2020 Olympic and Paralympic Games on the number of COVID-19 cases is underestimated. Third, measurement bias cannot be ruled out; some COVID-19 cases may have been missed due to a limited testing capacity and lack of comprehensive testing of asymptomatic individuals. The proportion of COVID-19 variants was estimated in the number of tests performed, and not in the total number of cases. Furthermore, the type of variant strain is unknown. Because COVID-19 cases caused by Delta strains were spreading very widely in Japan in the period before the opening of the Olympics, we wanted to at least adjust the proportion of the Delta variant; however, valid data were difficult to obtain. This measurement bias is a non-differential misclassification among all prefectures, leading to an underestimation of the COVID-19 cases. Fourth, there were unmeasured meteorological factors, such as the air temperature, relative humidity, and air quality, which significantly influence the spread and severity of pandemics [26]. In addition, we could not adjust for time-varying variables after the opening of the Tokyo 2020 Games and for specific or original policies in each prefecture for COVID-19 suppression. Alternatively, we adjusted for the number of COVID-19 cases over a long period before the event. Because this number was affected by a variety of unmeasured factors, this adjustment led to the minimization of the effect of unmeasured factors. Fifth, there was a lack of external validity. Our results cannot be applied to other countries and infectious diseases, because hosting the Tokyo 2020 Games under the COVID-19 pandemic is a unique situation. National characteristics, infectious diseases, and climatic conditions will change the results for other Olympics events and host cities. Therefore, these results should be generalized after careful consideration of the settings. Finally, although there were some limitations in this study, these limitations neither overestimated nor underestimated the evaluation of the impact of the Tokyo 2020 Games on the number of COVID-19 cases totally. Therefore, our results will draw attention to the potential impact of the Olympics on COVID-19 cases during the pandemic. Ultimately, a well-designed study with better statistical methods is necessary to clarify the causal inferences.

## 5. Conclusions

The Tokyo 2020 Olympic and Paralympic Games probably increased the number of COVID-19 cases, although it was difficult to adjust the influence of the fifth wave associated with the Delta variant. Further studies in different regions with more valid variables are needed to validate our findings. 

## Figures and Tables

**Figure 1 jpm-12-00209-f001:**
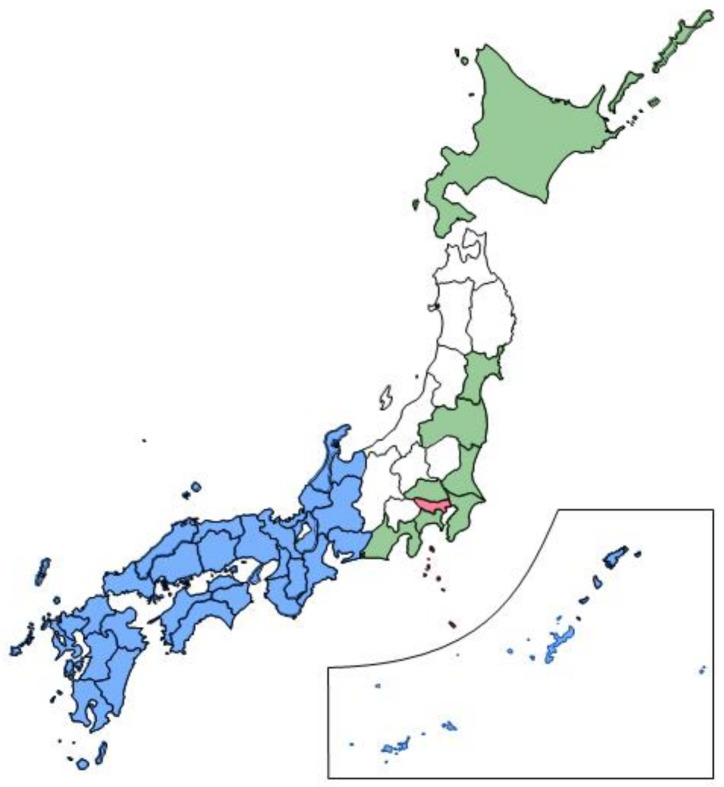
Map of the 47 prefectures in Japan. Red—Tokyo (host city); Green—prefectures where the Tokyo 2020 games were held; Blue—donor pool (29 prefectures west of Toyama, Gifu, and Aichi).

**Figure 2 jpm-12-00209-f002:**
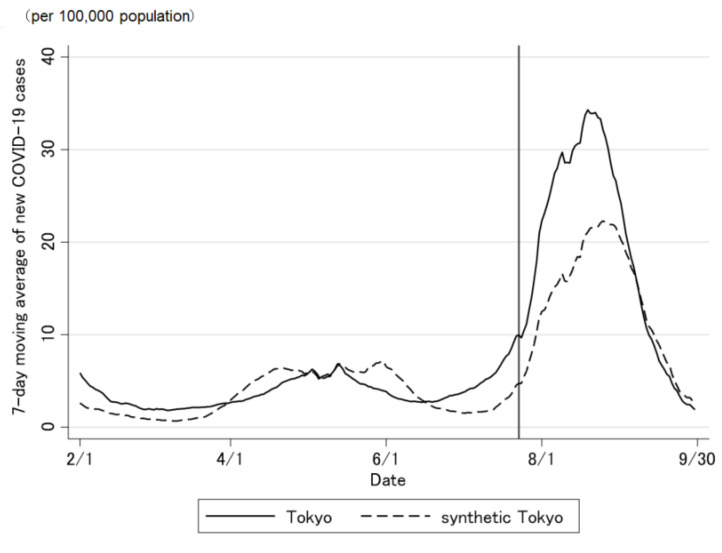
Trends in the daily number of newly confirmed COVID-19 cases in Tokyo and synthetic Tokyo, estimated using synthetic control weights. The vertical line on 23 July 2021 indicates the opening of The Tokyo 2020 Games.

**Figure 3 jpm-12-00209-f003:**
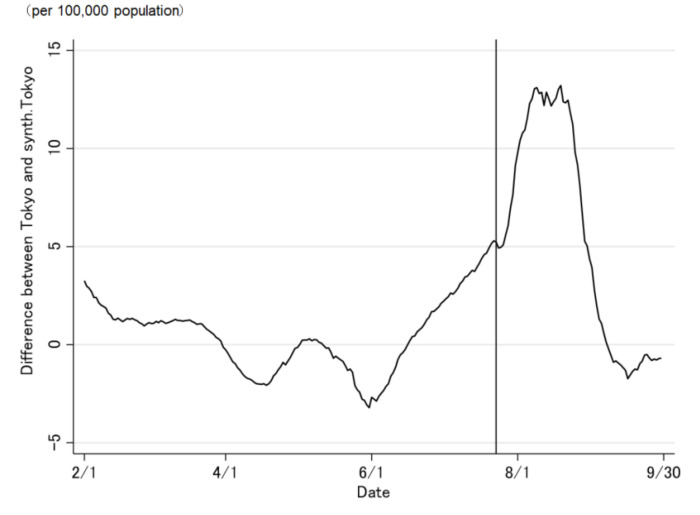
Difference in the daily number of newly confirmed COVID-19 cases between Tokyo and synthetic Tokyo, estimated using synthetic control weights.

**Figure 4 jpm-12-00209-f004:**
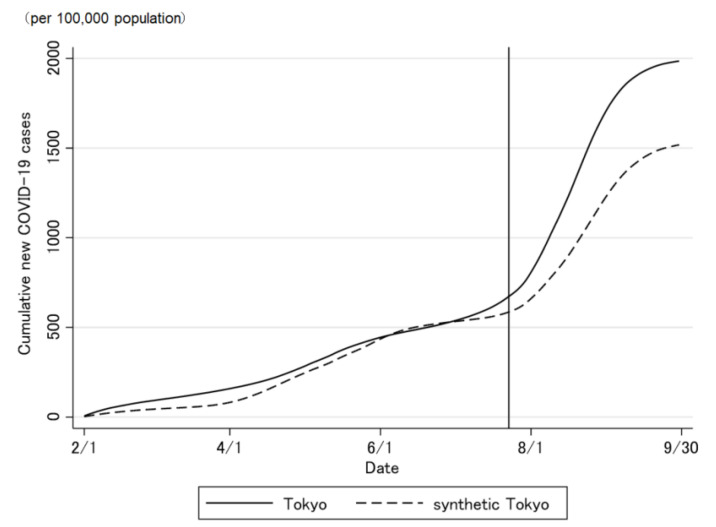
Trends of the cumulative daily number of newly confirmed COVID-19 cases in Tokyo and synthetic Tokyo, determined using synthetic control weights.

**Figure 5 jpm-12-00209-f005:**
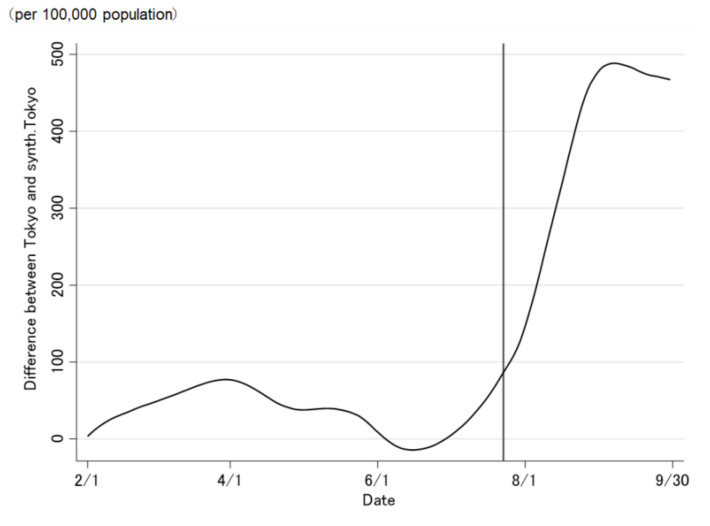
Difference in the cumulative daily number of newly confirmed COVID-19 cases between Tokyo and synthetic Tokyo, estimated using synthetic control weights.

**Figure 6 jpm-12-00209-f006:**
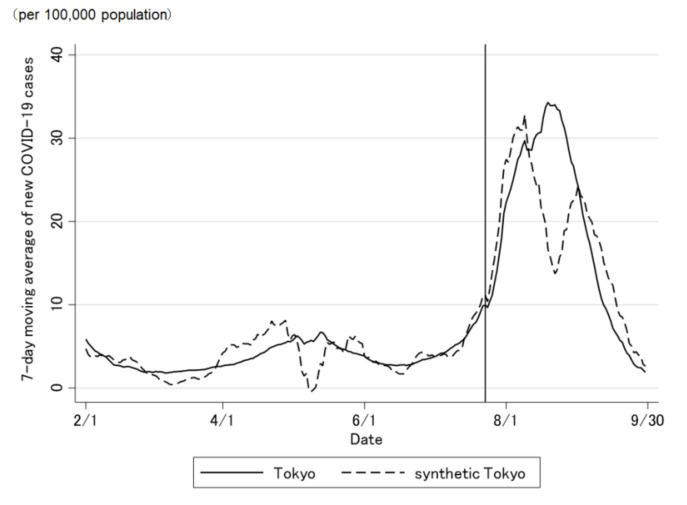
Trends of the daily number of newly confirmed COVID-19 cases in Tokyo and synthetic Tokyo, determined using regression weights.

**Figure 7 jpm-12-00209-f007:**
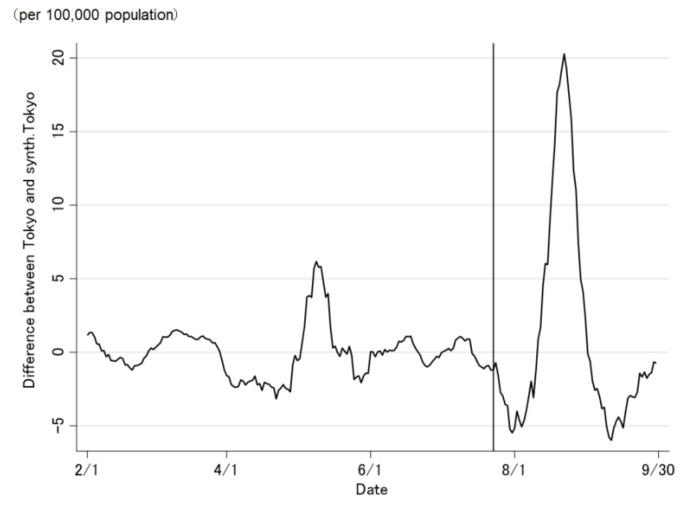
Difference in the daily number of newly confirmed COVID-19 cases between Tokyo and synthetic Tokyo, determined using regression weights.

**Figure 8 jpm-12-00209-f008:**
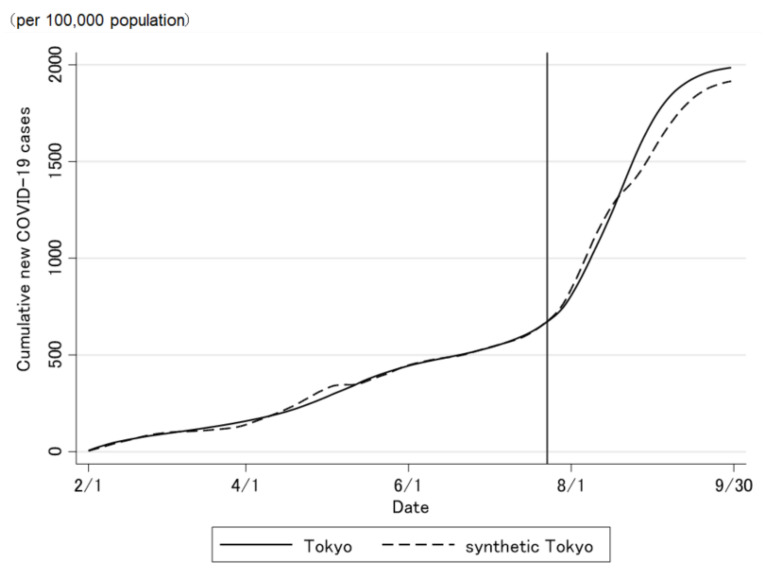
Trends of the cumulative daily number of newly confirmed COVID-19 cases in Tokyo and synthetic Tokyo, determined using regression weights.

**Figure 9 jpm-12-00209-f009:**
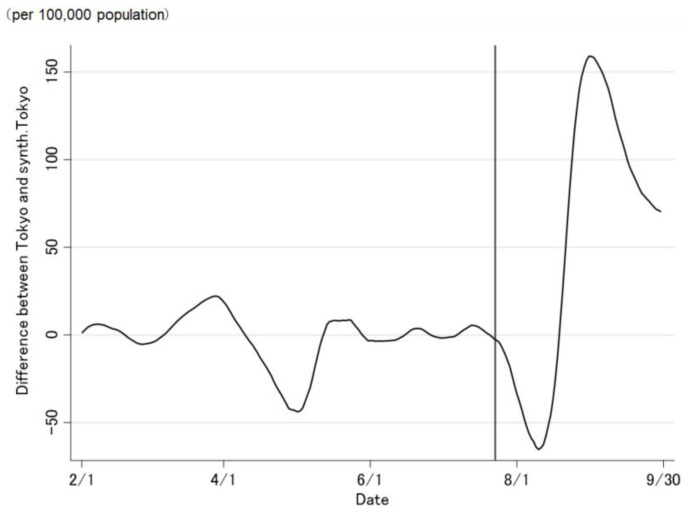
Difference in the cumulative daily number of newly confirmed COVID-19 cases between Tokyo and synthetic Tokyo, determined using regression weights.

**Figure 10 jpm-12-00209-f010:**
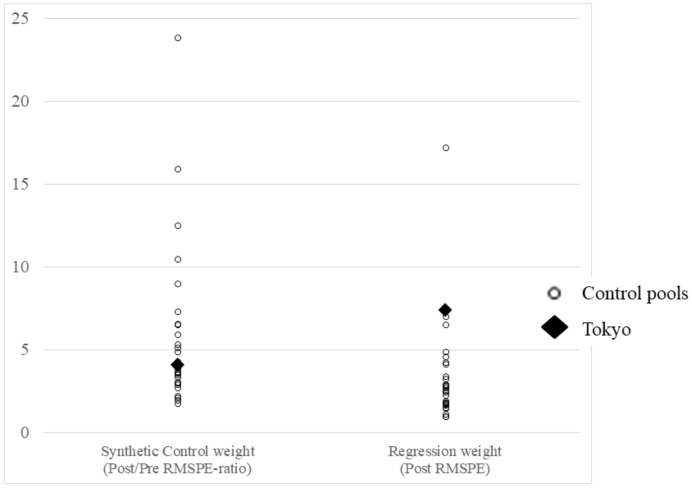
Total results of the synthetic control weights and regression weights. RMSPE—root mean square prediction error. Tokyo ranks 15th in the synthetic control weight analysis and second in the regression weight analysis.

**Table 1 jpm-12-00209-t001:** Synthetic control and regression weights for Tokyo.

Prefecture	Synthetic Control Weight	Regression Weight	Prefecture	Synthetic Control Weight	Regression Weight
Okayama	0	−0.729	Kumamoto	0	−0.171
Tokushima	0	−0.716	Shimane	0	−0.067
Miyazaki	0	−0.660	Aichi	0	−0.030
Kagoshima	0	−0.548	Okinawa	0.267	0.075
Shiga	0	−0.546	Kochi	0	0.218
Kagawa	0	−0.540	Nara	0	0.284
Hyogo	0	−0.493	Ehime	0	0.327
Wakayama	0	−0.492	Yamaguchi	0	0.398
Oita	0	−0.414	Mie	0	0.497
Hiroshima	0	−0.357	Kyoto	0	0.569
Saga	0	−0.328	Osaka	0.277	0.601
Gifu	0	−0.250	Fukui	0	0.888
Toyama	0	−0.233	Tottori	0	1.024
Nagasaki	0	−0.175	Fukuoka	0	1.392
				0.455	1.475

**Table 2 jpm-12-00209-t002:** Predictor means before the Tokyo 2020 Olympic and Paralympic Games.

	Tokyo	Synthetic Tokyo	Regression-Weighted Tokyo	Population-Weighted Average of the Control Pool
New case per 100,000 persons				
Mean between 1 February and 19 February 2021	3.71	1.78	3.71	1.18
Mean between 19 February and 8 March 2021	2.08	0.92	2.08	0.50
Mean between 8 March and 10 April 2021	2.40	2.06	2.40	1.14
Mean between 10 April and 13 May 2021	5.02	5.87	5.02	4.60
Mean between 13 May and 30 May 2021	5.05	6.42	5.05	4.81
Mean between 30 May and 15 June 2021	3.16	5.06	3.16	1.97
Mean between 15 June and 4 July 2021	3.26	1.97	3.26	0.77
Mean between 4 July and 22 July 2021	6.21	2.39	6.21	0.96
RMSPE of new COVID-19 cases	-	1.92	0.00	2.39
Related conditions				
Proportion of the population having received the second dose	41.3%	42.4%	41.3%	44.2%
Proportion of the COVID-19 variants in circulation	54.0%	18.9%	54.0%	27.1%
Proportion of people aged 15–64 years old	65.8%	59.5%	65.8%	58.3%
Proportion of people aged over 65 years old	23.1%	27.0%	23.1%	29.2%

RMSPE—root mean squared prediction error; COVID-19—coronavirus disease 2019.

## Data Availability

All the materials (The raw data and code) needed for the analyses are reposited on GitHub (https://github.com/sankyoh/Tokyo2020_COVID19) (accessed on 31 January 2022).

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
