# Peer review of "Causal Effect of the Tokyo 2020 Olympic and Paralympic Games on the Number of COVID-19 Cases under COVID-19 Pandemic: An Ecological Study Using the Synthetic Control Method"

_jpm, 2022, doi:10.3390/jpm12020209_

Round 1
Reviewer 1 Report
The authors estimated the causal effects of the Tokyo 2020 Games on the number of COVID-19 cases using the synthetic control method.
The authore recall a previous article in which is possible to find more details on the statistical analysis, that is:
Abadie, A.; Diamond, A.; Hainmueller, J. Synthetic Control Methods for Comparative Case Studies: Estimating the Effect of 365
California’s Tobacco Control Program. Journal of the American statistical Association 2010, 105, 493–505, doi: 366
10.1198/jasa.2009.ap08746
Overall, the introduction provides a summary on the topic and results and discussion are clear and show the main findings of the study. I do not have any suggestions to improve these sections.
Author Response
Response to Reviewer 1’s Comments:
Reviewer 1
Overall, the introduction provides a summary on the topic and results and discussion are clear and show the main findings of the study. I do not have any suggestions to improve these sections.
Response: Thank you for this positive comment.
Point 1: The author recall a previous article in which is possible to find more details on the statistical analysis, that is:
Abadie, A.; Diamond, A.; Hainmueller, J. Synthetic Control Methods for Comparative Case Studies: Estimating the Effect of 365
California’s Tobacco Control Program. Journal of the American statistical Association 2010, 105, 493–505, doi: 366
10.1198/jasa.2009.ap08746
Response 1: We have made the necessary change to the Statistical Analysis.

Reviewer 2 Report
This manuscript assessed the causal effect of Olympic Games on the spread of COVID using the synthetic control method and public data. The authors used the previous data and modeled a synthetic Tokyo without Olympic to analyze the impact of Olympic Games. The concept is interesting. However, their finding is somehow not less important as no results are statistically significant. The authors justified that such non-robustness is a result of some underestimated factors such as the spreading of Delta variant.
In general, although the concept/approach is interesting, the importance of the finding is below the bar of J. Personalized Medicine and hence I recommend a major revision for now and leave the choice of rejection to the editor.
In the revised manuscript, I hope the authors can address the critical problem of importance. As the public data regarding delta is pretty available now, they shall be able to include or decouple the impact of Delta spread fairly. This shall allow less biased justification on the goodness/quality of their models. In addition to the issue of importance, I have the following comments for readability/reproducibility/formatting.
- Please provide the workflow of modeling and statistical analysis as a figure and include the name, and version of the software/packages used in each step, and any pre-processing or data exclusion. The reviewer is not able to assess the information in some of the external SIs as it doesn’t allow anonymous access.
- Please remove the grey background in Fig. 2 – 9 as it does not meet the publisher’s figure guideline.
- The authors gave the synthetic control and regression weight of each prefecture at the beginning of the result section (page 4) – I had a hard time understanding the origin of these numbers. Please add a few sentences to explain the origin of tables at the first time of use. This shall allow better readability.
- Fig 3: please also report the difference as the % rather than the exact number. I guess this will result in a better fit between the two models.
Author Response
Response to Reviewer 2’s Comments:
Reviewer 2
Point 1: As the public data regarding delta is pretty available now, they shall be able to include or decouple the impact of Delta spread fairly.
Response 1: We searched for information on the number of COVID-19 cases due to the Delta variant in each prefecture in Japan around the opening of the Tokyo 2020 Games (July 23) again. However, we were unable to obtain these public data. Subsequently, we asked members of the National Institute of Infectious Diseases in Japan for the data. However, we still could not obtain these public data. We have determined that the data used in this study pertain to the L452R mutation (Delta variant), not all variants. We have, accordingly, made the following revision.
Page 3:
To determine the proportion of the variants of severe acute respiratory syndrome coronavirus 2 in circulation from July 19 to July 25, 2021, we used polymerase chain reaction (PCR) assays and the following formula: (number of cases of the L452R mutation [Delta variant] the PCR test was positive for)/(number of PCR tests performed for the L452R mutation).
Point 2: Please provide the workflow of modeling and statistical analysis as a figure and include the name, and version of the software/packages used in each step, and any pre-processing or data exclusion. The reviewer is not able to assess the information in some of the external SIs as it doesn’t allow anonymous access.
Response 2: We have created workflow figures for analysis. As the explanation is too verbose to include in the main text, we will publish it on Github. As you pointed out, anonymous access is not possible in some sites, but our Github page is public and completely accessible without logging in. The license is CC BY-SA 4.0, so anyone can reuse it. The following are the links: https://github.com/sankyoh/Tokyo2020_COVID19
and https://github.com/sankyoh/Tokyo2020_COVID19/blob/main/Workflow.md
For the analysis, we used Stata 17.0/MP4 with the user-driven command “synth” 0.0.7, Python 3.7.4, Numpy 1.18.1, Pandas 0.25.3, and Jupyter Notebook 6.0.2. The do files, py files, and ipynb file were created by Toshiharu Mitsuhashi for this study.
We have revised the workflow of modeling and statistical analysis as shown below. We have also added three supplemental workflow figures.
Page 4:
The analysis was performed using Stata/MP4 17.0 with the user-generated command “synth” 0.0.7 [20]. This calculation for regression weights was performed using Python 3.7.4, with NumPy 1.18.1, Pandas 0.25.3, and Jupyter Notebook 6.0.2.
Supplementary Figure 2: Workflow Figure 1.
Public data files (number of new infections, vaccinations, population, and cases of the Delta variant) by prefecture were converted from Excel files to Stata dta files. Three do files were used: crDataset.do, crDataset_population.do, and crDataset_variant.do. The converted Stata dta files were merged using the prefecture as the key. This was done by crMerge.do. For this file, we calculated the proportion of population under 15 years old, the proportion of population 15-64 years old, and the proportion of population that was vaccinated. This was done by crVariables.do. The file was named Dataset.dta and saved. Next, we divided the study period based on the peaks and troughs in the number of infections in Tokyo. In order to divide the periods, the boundary days were identified using anDetectPeek.do.
Supplementary Figure 3: Workflow Figure 2.
Normal SCM was performed with anSCM_simple2.do. To perform the SCM with regression weights, long-type data were converted to wide-type data. This was done using crConvert_LongtoWide.do. The converted file was output as an Excel file. This file was named df_wide_for_regwt.xlsx. SCM with regression weights was performed on this file. For this analysis, we used anRegWt_SCM2.py. Then, anRegWt_SCM2.ipynb was run to obtain the in-space placebo effect. These files output the weights for each prefecture.
Supplementary Figure 4: Workflow Figure 3.
Based on the weights, we drew a graph. We first merged the weights and the number of newly infected people in anDraw_graph_for_excel_newcase.do, and then created a data file for drawing the graph. The data file for drawing is named sum_result_for_graph.dta. Next, we used anDraw_graph_newcase.do to draw the graph.
Point 3: Please remove the grey background in Fig. 2 – 9 as it does not meet the publisher’s figure guideline.
Response 3: We have made the necessary changes to the figures.
Point 4: The authors gave the synthetic control and regression weight of each prefecture at the beginning of the result section (page 4) – I had a hard time understanding the origin of these numbers. Please add a few sentences to explain the origin of tables at the first time of use. This shall allow better readability.
Response 4: We have clarified the following under Materials and Methods.
Page 4:
Synthetic control weight:
We calculated the weight of the no-event (control) region from the donor pool in a way that the average pre-event trend and other selected variables were similar to the trend and characteristics of the event region (Tokyo). The weights were assigned based on the extent of the donor pool’s similarity to Tokyo prior to the Tokyo 2020 Games. The weights from the donor pool were chosen as follows: the weights for each prefecture ranged from 0 to 1, such that the weights for all control prefectures totaled 1.
Regression weight:
In addition, the SCM was performed by using regression weights [8]. This approach employs a linear combination for which the sum of the weights in the control pool is 1. Therefore, the weights were calculated as:
Wreg=X0'(X0X0')-1X1
Point 5: Fig 3: please also report the difference as the % rather than the exact number. I guess this will result in a better fit between the two models.
Response 5:In keeping with your comment, we can change Figure 3. However, we prefer using the difference in COVID-19 cases to facilitate a better understanding, based on previous SCM studies [4,6,7] and the original SCM study by Abadie et al.[2].
[2] Abadie, A.; Diamond, A.; Hainmueller, J. Synthetic control methods for comparative case studies: Estimating the effect of California’s tobacco control program. J. Am. Stat. Assoc. 2010, 105, 493–505.
[4] Born, B.; Dietrich, A.M.; Müller, G.J. The lockdown effect: A counterfactual for Sweden. PLoS One 2021, 16, e0249732.
[6] Zhu, P.; Tan, X. Is compulsory home quarantine less effective than centralized quarantine in controlling the COVID-19 outbreak? Evidence from Hong Kong. Sustain. Cities Soc. 2021, 74, 103222.
[7] Mills, M.C.; Rüttenauer, T. The effect of mandatory COVID-19 certificates on vaccine uptake: Synthetic-control modelling of six countries. Lancet Public Health 2022, 7, e15–e22.
[8] Abadie, A.; Diamond, A.; Hainmueller, J. Comparative politics and the synthetic control method. Am. J. Pol. Sci. 2015, 59, 495–510

Reviewer 3 Report
Although there were some limitations in this study, these limitations neither overestimated nor underestimated the evaluation of the impact of the Tokyo 2020 Games on the number of COVID-19 cases totally.
Therefore, their results will draw attention to the potential impact of the Olympics on COVID-19 cases during the pandemic.
Ultimately, a well-designed study with better statistical methods is necessary to clarify the causal inferences.
The Tokyo 2020 Olympic and Paralympic Games probably increased the number of COVID-19 cases, although it was difficult to adjust the influence of the fifth wave associated with the Delta variant. Further studies in different regions with more valid variables are needed to validate our findings

Author Response
Response to Reviewer 3’s Comments:
Reviewer 3
The Tokyo 2020 Olympic and Paralympic Games probably increased the number of COVID-19 cases, although it was difficult to adjust the influence of the fifth wave associated with the Delta variant. Further studies in different regions with more valid variables are needed to validate our findings
Response: We agree that more studies on this topic are needed.

Reviewer 4 Report
Review on the manuscript:
The Causal Effect of the Tokyo 2020 Olympic and Paralympic 2 Games on the Number of COVID-19 Cases under COVID-19 3 Pandemic: An Ecological Study Using the Synthetic Control 4 Method
This manuscript intends to determine whether or not the Olympic Games had an important impact on the COVID-19 infection process.
General comments
Some phases and paragraphs, despite being grammatically correct, are difficult to understand. This is perhaps due to the lack of some term meaning explanations. For example in Section 3.4, is not clear what the donor pool is. I have made an effort to encounter the meaning of the whole paragraph in Section 3.4 and I cannot find any reasonable meaning. Please rewrite this paragraph explaining what the donor pool is; Is the donor pool some sort of reference? What is it? Why attenuating the effects of the Olympic Games is important and what are the effects presumably being attenuated?
In line 127 : “Using the techniques described in a previous paper [2],”. Why not to mention the authors of this previous study? (Abadie, A.; Diamond, A.; Hainmueller, ) If you include their names and the names of their techniques, the reading becomes richer and more interesting. I suspect the method you use is the “Synthetic Control Methods“, but since you never mention it, I am, as reader, lost within the text and have very little perspective of your study. If this Synthetic Control Methods are the core of your procedure, I think it deserves an explanation within this manuscript…
After writing this I realize some explanation about the SCM is shown in the introduction. However, I insist there should be at lesat a little explanation of SMC. Especially in the Methods Section.
In Figure 4’s caption you include the word cumulative. I think this is wrong.
Final comments:
This manuscript presents a well written study based on applying the SMC method to decide whether or not there was a detectable impact on the contagious process after the Tokyo. Olympic Games. While the conclusions do not show a clear result of the study, this is arguably and well supported the result of applying the SMC method to this case. I think this manuscript deserves publication. It may improve by addressing the aspects mentioned above.
Author Response
Response to Reviewer 4’s Comments:
Reviewer 4
Final comments:
This manuscript presents a well written study based on applying the SMC method to decide whether or not there was a detectable impact on the contagious process after the Tokyo. Olympic Games. While the conclusions do not show a clear result of the study, this is arguably and well supported the result of applying the SMC method to this case. I think this manuscript deserves publication. It may improve by addressing the aspects mentioned above.
Response: Thank you for the positive comments on our manuscript.
General comments
Point 1: Some phases and paragraphs, despite being grammatically correct, are difficult to understand. This is perhaps due to the lack of some term meaning explanations. For example in Section 3.4, is not clear what the donor pool is. I have made an effort to encounter the meaning of the whole paragraph in Section 3.4 and I cannot find any reasonable meaning. Please rewrite this paragraph explaining what the donor pool is; Is the donor pool some sort of reference? What is it? Why attenuating the effects of the Olympic Games is important and what are the effects presumably being attenuated?
Response 1: Alberto Abadie, who developed the synthetic control method, recently described that it is based on the observation that a combination of units in “the donor pool” may approximate the characteristics of the affected unit substantially better than any unaffected unit alone [27]. A synthetic control is defined as a weighted average of the units in the donor pool [27]. In this study, the synthetic Tokyo was created as a combination of three prefectures, namely Ishikawa, Osaka, and Okinawa (weights: 0.455, 0.277, and 0.267, respectively), which reproduced the COVID-19 case trends in Tokyo before the Tokyo 2020 Games.
Attenuating the effects of the Olympic Games will reduce the impact of large-scale events during a pandemic, thereby achieving both the economic benefits of the event and the prevention of the spread of infectious diseases. If infection is reduced, the socioeconomic benefits of the event will not be lost. We have made the following revisions.
Page 3:
Abadie et al. defined donor pools as regions with similar characteristics to the region exposed to the event of interest [2]. We constructed the counterfactual “synthetic Tokyo” from a unit in the donor pool of prefectures in Japan.
Page 11:
The results of this study can offer important insights for future decisions on hosting the Olympics and international mass gathering events and framing policies during pandemics, considering the socioeconomic benefits and burdens.
[2] Abadie, A.; Diamond, A.; Hainmueller, J. Synthetic control methods for comparative case studies: Estimating the effect of California’s tobacco control program. J. Am. Stat. Assoc. 2010, 105, 493–505
[27] Abadie, A. Using synthetic controls: Feasibility, data requirements, and methodological aspects J. Econ. Lit. 2021, 59, 391–425.
Point 2: In line 127 : “Using the techniques described in a previous paper [2],”. Why not to mention the authors of this previous study? (Abadie, A.; Diamond, A.; Hainmueller, ) If you include their names and the names of their techniques, the reading becomes richer and more interesting. I suspect the method you use is the “Synthetic Control Methods“, but since you never mention it, I am, as reader, lost within the text and have very little perspective of your study. If this Synthetic Control Methods are the core of your procedure, I think it deserves an explanation within this manuscript…
Response 2: We have revised the related sentences as shown below.
Page 4:
We evaluated the causal effect of the Tokyo 2020 Games on the number of COVID-19 cases using the SCM, as proposed by Abadie et al. [2]. We calculated the weight of the no-event (control) region from the donor pool in a way that the average pre-event trend and other selected variables were similar to the trend and characteristics of the event region (Tokyo). The weights were assigned based on the extent of the donor pool’s similarity to Tokyo prior to the Tokyo 2020 Games. The weights from the donor pool were chosen as follows: the weights for each prefecture ranged from 0 to 1, such that the weights for all control prefectures totaled 1.
[2] Abadie, A.; Diamond, A.; Hainmueller, J. Synthetic control methods for comparative case studies: Estimating the effect of California’s tobacco control program. J. Am. Stat. Assoc. 2010, 105, 493–505.
Point 3: After writing this I realized some explanation about the SCM is shown in the introduction. However, I insist there should be at lesat a little explanation of SCM. Especially in the Methods Section.
Response 3: We have revised the explanation of the SCM under Materials and Methods. In section 3.7, we have added a note on weight selection.
Page 4:
The weights for the donor pool were chosen as follows. X1 be the (12 × 1) vector representing the variables of the Kanto region before the intervention; X0 be the (12 × 29) matrix signifying the variables of the control pool before the intervention; and W be the (29 × 1) vector denoting the weight of the control pool. The weight W = (w1, w2, ..., w29) was selected to minimize ||X1-X0W||, subject to 0 ≦ wj ≦ 1 for all j, and w1 + w2 + … + w29 = 1.
…This approach employs a linear combination for which the sum of the weights in the control pool is 1. Therefore, the weights were calculated as:
Wreg = X0'(X0X0')-1X1
Point 4: In Figure 4’s caption you include the word cumulative. I think this is wrong.
Response 4: We apologize for the confusion; the figure you are referring to was incorrectly labeled “4.” The actual Figure 4 shows the trends of the cumulative daily number.

Round 2
Reviewer 2 Report
The authors made some revisions based on my comments – which I really appreciate. However, the novelty/scientific soundness of the article is still below the bar of publication. As the author claimed, this study already has many limitations, so it is important to be able to quantify the impact of Delta to show the correctness of their modeling/analysis.
Similar to my previous recommendation, I will leave the decision to the editor. If you feel the current novelty/scientific soundness is okay, this MS can be accepted after a minor formatting/language editing. Otherwise, I recommend the authors wait for the official data release regarding Delta strain and consider that to fully explain the goodness of their model.